# Dynamics of Anthropogenic Wildfire on Babeldaob Island (Palau) as Revealed by Fire History

**Julian Dendy** [1,*] **, Dino Mesubed** [2] **, Patrick L. Colin** [1] **, Christian P. Giardina** [3] **, Susan Cordell** [3] **, Tarita Holm** [4] **and Amanda Uowolo** [3]

1   Coral Reef Research Foundation, P.O. Box 1765, Koror 96940, Palau; crrfpalau@gmail.com
2   Bureau of Agriculture, Palau Forestry, Ngerelmud 96940, Palau; boagri@palaunet.com
3   Institute of Pacific Islands Forestry, Pacific Southwest Research Station, USDA Forest Service, 60 Nowelo St., Hilo, HI 96720, USA; christian.p.giardina@usda.gov (C.P.G.); susan.cordell@usda.gov (S.C.); amanda.l.uowolo@usda.gov (A.U.)
4   Russell Square, 10 Thornhaugh Street, University of London SOAS, London WC1H 0XG, UK; taritaholm@gmail.com
*   Correspondence: jdendy@hawaii.edu; Tel.: +680-488-5255

**Abstract:** Wildfire is an understudied threat to biodiversity in many tropical landscapes, including island nations of the Pacific, such as The Republic of Palau—a global biodiversity hotspot with ridge-to-reef resources. Fires are known to occur on Palau's main island of Babeldaob, where they can result in increased erosion rates and sediment delivery to near-shore areas with impacts to streams and coral reefs. Fire-adapted native plant species are found in savanna habitats, but fires often extend into adjacent forest areas where they kill overstory trees. To assess this serious biodiversity and human health threat, we mapped wildland fires on Babeldaob Island using ground-based surveys and aerial photographs between 2012 and 2015, and satellite imagery between 2012 and 2021. Data on causal factors, vegetation type, and the presence of invasive species were collected between 2012 and 2015, with hunting, arson, and agricultural clearing being the principal causes of ignitions. Wildfires occurred in all months and in all 10 states of Babeldaob, and both numbers of wildfires and total burned area were substantially greater during dry seasons, with the highest totals occurring in the one El Niño drought year in our record. Overall, wildfires appear to have a minor impact on forest vegetation because they are largely confined to savanna systems, but rainfall on burned savanna is a major cause of erosion and the sedimentation of streams and near-shore habitats.

**Keywords:** wildfire; Micronesia; forest; savanna; hunting; agriculture; GIS; GPS; remote sensing

## 1. Introduction

Wildfires are a growing threat in the tropics, with diverse impacts to ecology and human health [1,2]. These wildfires are often human-caused and typically result from activities in disturbed settings, such as slash and burn clearings of secondary forest or agricultural waste management. These intentional fires can escape and move into forests, causing tree mortality and significantly reducing forest cover, often at large scales [1,3–5]. Some escaped wildfires are driven by invasive grasses that encroach into forest edges [5,6]. Droughts, which are often associated with El Niño events, result in the most severe and extensive fires [2]. These fires have negative consequences for biodiversity [1,2,7–9] and ecosystem functioning [10–14], with synergies among wildfire, invasive species, and climate change being particularly disruptive [5,15–17].

As climate change in tropical islands intensifies the weather conditions that increase the risk of wildfire [2,16], public safety and natural resource agencies are increasingly investing in wildfire prevention, fuel mitigation, wildfire suppression, and the post-fire restoration of burned areas, all of which benefit from accurate wildfire histories. Impacts of anthropogenic wildfire on small tropical islands are accentuated because they are highly vulnerable to

climate change [16,18], have a high proportion of total land area potentially affected by fire [15], and have compact watersheds which potentially exacerbate sedimentation to near-shore areas. However, for many tropical island nations, wildfire histories do not exist, and underlying causes are poorly documented.

Wildfire documentation efforts in much of the tropics have historically relied on visual estimates of burned area, which have the benefit of sometimes including relatively long fire histories but are imprecise in terms of annual wildfire frequency and burned area metrics. More recently, remote sensing techniques incorporating satellite or aerial imagery have been widely applied to estimate burned area and other wildfire metrics, with high-to-moderate-resolution imagery being most appropriately used at local scales [19]. Various burned area products and spectral indices have been developed and used to map burned areas independently of active fire products, and the Normalized Burn Ratio (NBR) has demonstrated utility in grassland habitats including on small tropical islands [19,20]. Unfortunately, although these current methods are suitable for most research purposes, they are prone to high rates of commission and omission errors; thus, research into methods that increase burned area accuracy is still needed [20,21]. In addition, although remote sensing is very efficient compared with ground-based monitoring, it cannot provide fine-scale observations of the vegetation that was burned, the presence of invasive species, the depth of forest edge penetrated by fire, or probable reasons for ignitions. As such, utilizing and comparing a combination of wildfire documentation methods, including field-based methods and the use of high-resolution satellite imagery, can be useful in more accurately quantifying the impacts of wildfires on tropical island ecosystems.

Micronesia is a geographic region in the western Pacific Ocean, where anthropogenic wildfires pose substantial ecological threats, but formal documentation of wildfire history is lacking. Micronesia's nation states, particularly in the westernmost part, are characterized by many small tropical islands that experience 3–4-month dry seasons of roughly $\geq 30\%$ lower-than-average precipitation, resulting in a seasonal pattern of human caused wildfires and the presence of substantial savanna habitat on all volcanic islands in the region [22]. This includes Babeldaob Island, Republic of Palau, which is the second largest island in Micronesia and makes up over three-quarters of Palau's land area. Palau's national guiding documents have called for an increased quality and extent of wildfire monitoring data on Babeldaob, including fire size, location, and cause, to better inform natural resource management and public safety planning [23,24]. This study addresses that data request, and reports on efforts between 2012 and 2021 to quantify the spatial extent, temporal variability, ecological characteristics, and social drivers of fire on Babeldaob Island from 2012 to 2021. Our study questions were: (1) What is the annual area burned; (2) What are the spatial and temporal patterns of wildfires; (3) What is the level of wildfire threat to forest resources; and (4) What are the principal causes of wildfire ignitions on Babeldaob Island?

In Section 2, we describe the study area including vegetation cover and savanna characteristics. In Section 3, we describe the field-based and remote sensing methods used to characterize wildfires on Babeldaob Island (2012–2021). In Section 4, we describe the geospatial and temporal wildfire trends on Babeldaob during the study period, including invasive species associated with burned areas. In Section 5, we discuss the benefits and limitations of our relatively low-tech, low-cost approach, and further elaborate on the geospatial trends in wildfires on the island and potential climate-related and anthropogenic impacts on the landscape and near shore. Finally, Section 6 concludes with a summary of answers to our study questions and an appraisal of the methodology.

## 2. Study Area

The Republic of Palau is a tropical island archipelago situated at the southwestern edge of Micronesia in the northwestern Pacific Ocean at a latitude of about 7° north and longitude of 134° east (Figure 1). The climate of Palau is hot and humid, with the daily average temperature maximum of 31.2 °C varying by less than 1.1 °C throughout the year [25]. Annual rainfall averages about 3734 mm but monthly averages from February

to April are lower (221 mm) than the annual monthly average (310 mm). The nation contains hundreds of islands, mostly of volcanic or raised limestone origin, of which about a dozen are inhabited. Palau contains the largest expanse of native lowland forest and the highest number of plant species including endemics (20% are unique to Palau) in Micronesia [26–29].

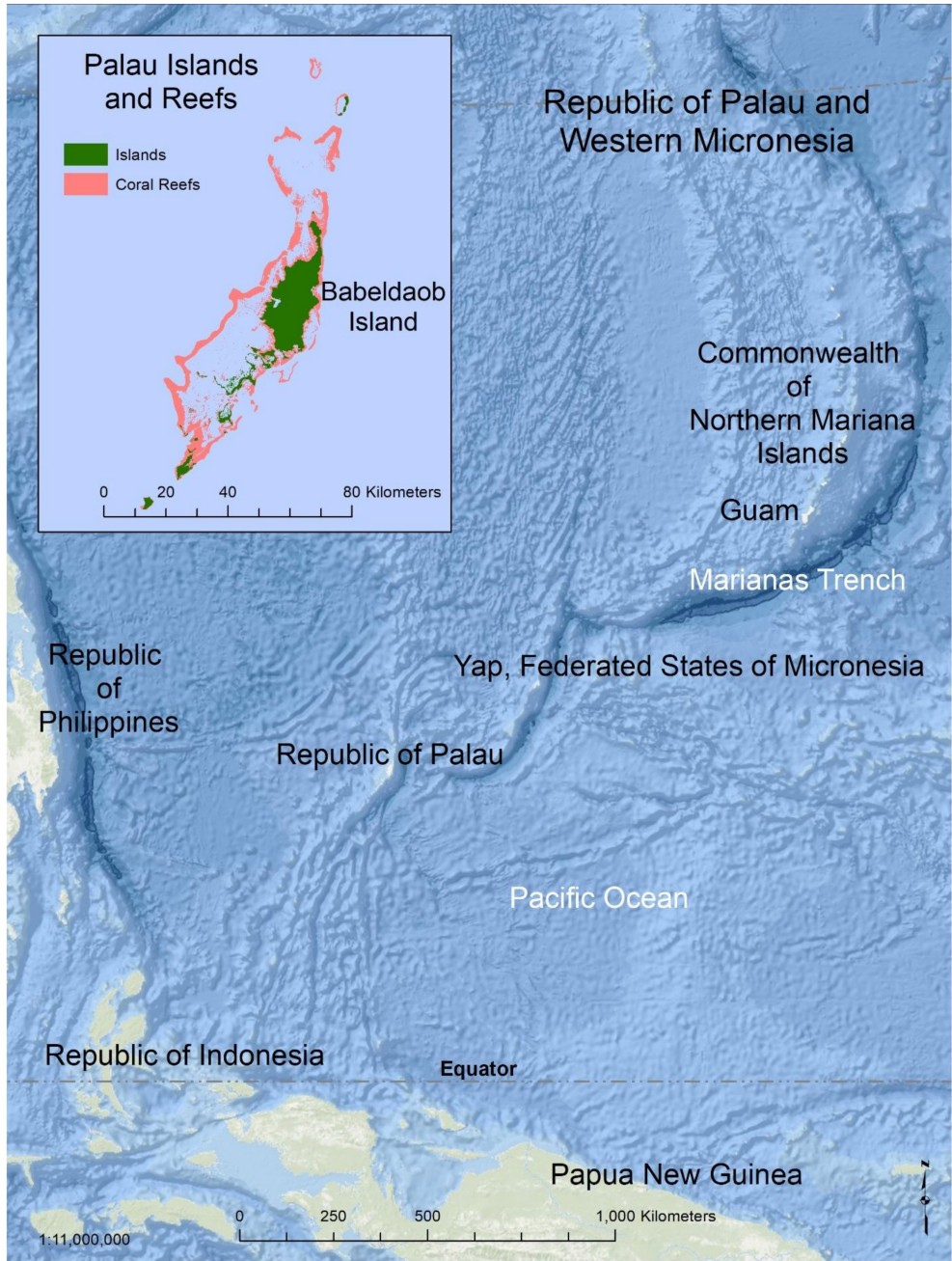

**Figure 1.** Location of the Republic of Palau and western Micronesia region within the Pacific Ocean.

Our study site, the volcanic high island of Babeldaob, comprises ~365 sq. km (78%) of the total ~466 sq. km of Palau's land area, contains ten states, and is home to about 5000 human residents, representing 30% of the nation's population [30]. The island has a rolling, rugged landscape, and at least 46 km$^2$ of the terrain has been shaped into immense earth architecture over three millennia of human history [31]. Wildfires appear to have arrived in Palau with early inhabitants, at least 3000 years ago, as indicated by the appearance in paleo records of charcoal, taro, and increases in various grasses [32,33].

These human-caused fires likely created and maintained savanna habitats on Babeldaob, and over centuries, the relative distribution of forest and savanna has fluctuated [32].

Babeldaob contains most of the volcanic rainforest, stream, and savanna habitats in the country, with savanna covering about 18% of the island [34]. A vegetation survey of Palau lists five subtypes of savanna on Babeldaob—bare, grassland, fern lands, shrubs, and abandoned agriculture [34]—and 179 species are thought to occur primarily in savanna [35], although all savanna species co-occur in forest habitats [28]. The island is surrounded by near-shore fringe reef ecosystems as well as an extensive barrier reef that support globally significant coral and fish biodiversity [36]. Babeldaob also has freshwater resources of considerable ecological and municipal value, including the largest freshwater lake (Ngardok) and the largest estuary (Ngeremeduu Bay) in Micronesia. Wildfires expose soil surfaces to rain; therefore, fires increase erosion rates and sediment delivery to streams, mangrove swamps, and near-shore ecosystems, including sea grass beds and coral reefs. Wildfire management, therefore, plays an important ecological role in marine conservation by sustaining ecosystems that are economically and culturally vital to the communities of Palau, as in other areas of Micronesia [36–39].

## 3. Materials and Methods

### 3.1. Methods Introduction

When this study began in 2012, our primary goal was to estimate the annual area of Babeldaob Island burned by wildfires, and at that time, two methods of quantifying burned area were known and available to us: ground-based surveys and aerial photos. Ground-based surveys of wildfires were primarily used to estimate burned area from 2012 to 2015 and supplemented with bi-annual aerial photo collections of the island. In 2016, an online high-resolution satellite image service became available, which we used to document burned areas from 2016 to 2021. We also used this approach to validate, and where needed, correct or supplement mapped burned areas from 2012 to 2015 (Table 1 and Figure 2).

**Table 1.** Number (and percentage) of wildfires and percentage of total burned area by year and documentation method on Babeldaob Island, Palau, for 2012–2015. All fires after 2015 were mapped using satellite imagery.

| Fire Year | GPS | Aerial Photos | Satellite Imagery |
|---|---|---|---|
| 2012 | 12 (6.9%) | 43 (24.7%) | 119 (68.4%) |
| 2013 | 130 (60.7%) | 21 (9.8%) | 63 (29.4%) |
| 2014 | 39 (42.9%) | 11 (12.1%) | 41 (45.1%) |
| 2015 | 106 (32.4%) | 22 (6.7%) | 199 (60.9%) |
| **% of Burned Area** | **GPS** | **Aerial Photos** | **Satellite Imagery** |
| 2012 | 2.5% | 25.2% | 72.3% |
| 2013 | 72.5% | 9.5% | 18% |
| 2014 | 23.9% | 9.9% | 66.2% |
| 2015 | 33.7% | 10.2% | 56.4% |

Garmin Map 60 Global Positioning System (GPS) units and a light aircraft equipped with GPS-enabled cameras were used to map wildfire boundaries on Babeldaob Island, Palau, from 2012 to 2015. In the field, smaller fires (<4 ha) were mapped by a single crew member, whereas larger fires were mapped by a pair of field crew members walking in opposite directions. Fires were mapped a few days to up to a year after the burn. Fire severity was not assessed. Each burned area was photographed, and observations were documented by the vegetation type burned, if and how far a fire extended into the forest edge, the presence and identity of invasive species, and if possible, the cause of the fire. Residents in, adjacent to, or near burned areas were interviewed to help identify the

potential reason(s) for ignition(s) that caused a fire. Additional observations of conditions surrounding the fire were used to assign the cause if residents were unavailable.

**Burned Area Documentation**
- Collection of burned area data on Babeldaob Island, Palau (2012–2021);
- Handheld GPS with field observations, aerial photos from light aircraft with metadata (2012–2015);
- Satellite image service (DGEV, 2016–2021).

**Data Processing**
- In ArcMap, manually digitize GPS tracks and enter field data into shapefile attribute tables, georeference aerial photos, and manually digitize burn scars;
- In DGEV, create 5 x 5 km$^2$ grid for assessment, visually scan each cell with each image, use advanced search when a burned area is detected, if more than one image is available, select one based on the clearest view, closest date to fire, lowest off nadir angle, WV3 before WV2 before GEO1 before WV1, draw a rectangle to clip the burned area, name and dowload the image file, manually digitize in ArcMap, add date if possible from temporal progression in imagery, with an estimated accuracy of ±4 days.

**Data Analysis and Summary**
- Download >50% cloud-free archival satellite imagery (2012–2015), compare with field-based and aerial-photo-based burned area polygons and estimated dates, use satellite image to manually digitize burned area and update shapefile when previously missed, closer to fire date, a clearer image or otherwise obviously more accurate than polygon;
- Merge all annual shapefiles including updated shapefiles (2102–2015), use Babeldaob state boundary, road, stream, mangrove, coastline, protected area and vegetation cover shapefiles, and select and clip tools with the merged fire shapefile to create geospatial summaries of burned areas, use the intersect tool to assess repeated burns.

**Figure 2.** Flow chart of steps to document a wildfire history on a small landscape.

Identified reasons for ignitions included: (i) general vegetation clearing, (ii) burning of brush piles, (iii) land preparation for agricultural activities, (iv) land preparation for house construction, (v) hunting/gathering, (vi) arson, and (vii) government activity. Category v was assigned if evidence of animal remains (bird feathers) or an adjacent marked hunting trail were observed, or if the fire was >500 m from the nearest roadway. Hunters and gatherers on Babeldaob were generally seeking Nicobar pigeon (*Caloenas nicobarica*, Linaeus, 1758), Micronesian imperial pigeon (*Ducula oceanica*, Desmarest, 1826), Palau flying fox (*Pteropus pelewensis*, K. Andersen, 1908), feral chickens (*Gallus gallus*, Linaeus, 1758), mangrove crab (*Scylla* sp.), or betel nut (*Areca catechu*, L.). Category vi was assigned when burned areas were near roads and evidence of other causes was not observed. Category vii was usually identified by an area resident and involved public works such as road-side grass cutting or utility-maintenance-related vegetation clearing or cutting.

In 2016, the Digital Globe EV Webhosting™ (DGEV) imagery service became available and was used to document burned areas on Babeldaob Island, Palau, from 2016 to 2021, to supplement the 2014 and 2015 burned area datasets and to validate, correct, and supplement datasets from 2012 to 2015. This service provides a near-complete image set of the island approximately every three weeks, and includes World View 1, 2 and 3, and GEO1 image sources with sub-meter resolutions.

*3.2. Data Introduction*

When a burned area was surveyed with handheld GPS, track logs were uploaded as a text file to a computer with DNR Garmin and converted into an Excel file. ArcMap 10.1™ was used for visualizing and manually converting track logs into burned area polygons, with a Quickbird satellite mosaic image (2004–2006) used as the island base map. GPS unit

accuracy was constrained by the instrument ($\pm 7.5$ m), and a full buffer (7.5 m) of burned area polygons was used to generate the accuracy of the estimated total area burned.

A free software program, BR EXIF Extractor, was used to extract the metadata and GPS coordinates from aerial photographs into a text file for each photograph collection. The metadata included photo ID, date, time, altitude, latitude, longitude, and GPS date and time. ArcMap 10.1 was used to visualize the flight path corresponding to the aerial photos of interest. Photos were of very high quality and showed burned areas in great detail. They were visually examined and selected for obvious fires and georeferenced to the base Quickbird satellite image of Babeldaob. The accuracy of georeferenced aerial photos was estimated to be that of the base satellite image, or $\pm 5$ m. Fire scars in aerial photos were sometimes still visible from events that occurred 3 or more years previously; thus, satellite imagery was used to confirm fire years in those cases.

In the DGEV imagery service, a $5 \times 5$ km$^2$ cell grid was used to ensure consistency of fire monitoring across the island and across years. Each available satellite image for each grid cell was zoomed to 2.4 m per pixel (level 15) in the online image viewer in all years, except 2016, when the zoom level was set at 1.2 m per pixel (level 16). Images were visually scanned for evidence of burned areas, clipped, saved as orthorectified geoTIFF files, and used to manually digitize burned areas in ArcMap. A fire date was assigned where possible, with an estimated accuracy of $\pm 4$ days. Satellite images were georeferenced and orthorectified, but high off-nadir angle values in some images resulted in a mean accuracy of $\pm 5$ m. Full buffers (5 m) were used to estimate the accuracy of total burned area documented by aerial photos and satellite images.

Burned area data documented from all sources were compiled into a single geospatial layer containing information on all wildfires on Babeldaob observed from 2012 to 2021, and descriptive statistics were developed to perform comparisons across years and states without considerations of the collection method. Additional geospatial layers were used in ArcMap to analyze spatial and temporal patterns of Babeldaob wildfires, including a 2014 vegetation map of Babeldaob [40], and shapefiles of island roads, streams, mangroves, coastline, and protected areas [41] (data citation).

## 4. Results

Between 2012 and 2021, at least one fire occurred in every state every year on Babeldaob Island, Palau, except in 2021 (Table 2 and Figure 3). The annual area burned varied eighteen-fold across the study (38–679 ha), with 153 fires per year (95% CI:125–180) burning 302 ha (95% CI:238–366), or 0.8% of the total island area (95% CI:0.7–1.0%) (Table 2). Precipitation was a major factor in determining wildfire occurrence, as 52% of fires occurred in the three dry season months of February, March, and April, whereas only 11% occurred in the three wet season months of October, November, and December. Overall, mean monthly rainfall was inversely correlated with the mean number of monthly fires (Figure 4). The El Niño drought in 2015 was the most severe fire year documented in our history, with more than twice the mean total and proportional annual area burned. The mapping and documentation of burned areas on Babeldaob from 2012 to 2021 was informed by three methodologies. Mapping with satellite imagery was the most thorough method, with an estimated burned area accuracy of 2193 $\pm$ 310 ha. Bi-annual aerial photo mapping had a comparable but lower burned area accuracy of 180 $\pm$ 37 ha and provided high-quality detailed imagery. Handheld GPS had the lowest estimated burned area accuracy (642 $\pm$ 175 ha) and informed the documentation of wildfire characteristics on the ground.

Vegetation on Babeldaob was differentially impacted by wildfires, with savanna vegetation types (bare, grassland, fern lands, shrubs, and abandoned agriculture) supporting the primary combusted material, with 36% (1760 ha) of the island's savanna cover burning over ten years. Forest was much less affected proportionally than savanna by wildfires, with only 1% (303 ha) burning during the study. Burning was highly variable within and nearby terrestrial protected areas and ranged as high as 11% of total burned area per year but appears to have declined over the course of the study (Table 3).

The great majority (88%) of fires were small (<4 ha) and comprised a very large proportion of total burned area (83%) during the study period, whereas the largest two fires alone made up about 3% of the total burned area (Table 4 and Figure 5). By cause, wildfires started by hunter–gatherers were the largest (mean = 3.3 ha; 95% CI: 2.99–3.60), whereas those associated with house building preparation were the smallest (mean = 1.2 ha 95% CI: 0.62–1.82), with other categories being of intermediate sizes (Table 5).

**Table 2.** Number of wildfires on Babeldaob Island, Palau, burned by year from 2012 to 2021, size range, mean area burned (ha), sum of area burned (ha) and % of total island area burned.

| Fire Year | # of Fires | Size Range (ha) | Mean Area (ha) | Sum of Area (Ha) | % Island Area |
|---|---|---|---|---|---|
| 2012 | 174 | <0.1–53 | 3.0 | 515 | 1.4% |
| 2013 | 214 | <0.1–31 | 2.3 | 500 | 1.4% |
| 2014 | 91 | <0.1–17 | 1.8 | 165 | 0.5% |
| 2015 | 327 | <0.1–36 | 2.1 | 679 | 1.9% |
| 2016 | 239 | <0.1–13 | 0.9 | 226 | 0.6% |
| 2017 | 110 | <0.1–9 | 0.9 | 103 | 0.3% |
| 2018 | 132 | <0.1–24 | 2.1 | 280 | 0.8% |
| 2019 | 100 | <0.1–29 | 2.2 | 220 | 0.6% |
| 2020 | 109 | <0.1–73 | 2.6 | 289 | 0.8% |
| 2021 | 29 | <0.1–8 | 1.2 | 38 | 0.1% |
| Total | 1525 | <0.1–33 | 19.3 | 3016 | 8.3% |
| Mean | 153 | <0.1–26 | 1.9 | 302 | 0.8% |

**Table 3.** Number, percentage, mean area (ha), sum of area (ha) and % of total area of wildfires inside and within 50 m of protected areas (PA's) on Babeldaob Island, Palau, per year from 2012 to 2021.

| Protected Area (PA) Fires | 2012 | 2013 | 2014 | 2015 | 2016 | 2017 | 2018 | 2019 | 2020 | 2021 |
|---|---|---|---|---|---|---|---|---|---|---|
| Fires Inside PAs | 12 | 16 | 1 | 15 | 13 | 2 | 7 | 4 | 2 | 0 |
| % of All Fires | 6.9% | 7.5% | 1.1% | 4.6% | 5.4% | 1.8% | 5.3% | 4.0% | 1.8% | 0.0% |
| Mean Area (ha) | 4.7 | 1.5 | 0.5 | 1.4 | 0.7 | 2.2 | 2.9 | 0.7 | 4.3 | 0.0 |
| Sum Area (ha) | 56.7 | 24.2 | 0.5 | 21.5 | 8.8 | 4.4 | 20.0 | 3.0 | 8.6 | 0.0 |
| % of Total Area | 11.0% | 8.4% | 5.0% | 7.1% | 9.7% | 4.3% | 7.4% | 1.3% | 3.0% | 0.0% |
| Fires Within 50 m of PAs | 23 | 34 | 10 | 40 | 33 | 4 | 11 | 10 | 9 | 2 |
| % of All Fires | 13.2% | 15.9% | 11.0% | 12.2% | 13.8% | 3.6% | 8.3% | 10.0% | 8.3% | 6.9% |
| Mean Area (ha) | 4.5 | 35.6 | 1.2 | 2.8 | 1.0 | 2.4 | 2.2 | 2.6 | 3.9 | 0.6 |
| Sum Area (ha) | 102.5 | 120.7 | 12.3 | 11.2 | 31.9 | 9.5 | 23.9 | 25.9 | 35.0 | 1.1 |
| % of Total Area | 19.9% | 24.1% | 7.4% | 1.7% | 14.1% | 9.2% | 8.5% | 11.8% | 12.1% | 2.9% |

**Table 4.** Number of wildfires, mean number per year, burned area (ha), and mean area burned per year by U.S. National Wildfire Coordinating Group size class (<0.1 ha, 0.1 to <4 ha, 4 to <40 ha, >40 ha), on Babeldaob Island, Palau, from 2012 to 2021.

| Fire Size Class (ha) | # of Fires | Fires Per Year | Burned Area (ha) | Area Burned Per Year (ha) |
|---|---|---|---|---|
| <0.1 ha | 96 | 10.7 | 60.2 | 6.0 |
| 0.1 to <4 ha | 1242 | 138.0 | 2443.4 | 244.3 |
| 4 to <40 ha | 185 | 20.6 | 421.7 | 42.2 |
| >40 ha | 2 | 0.2 | 90.3 | 9.0 |

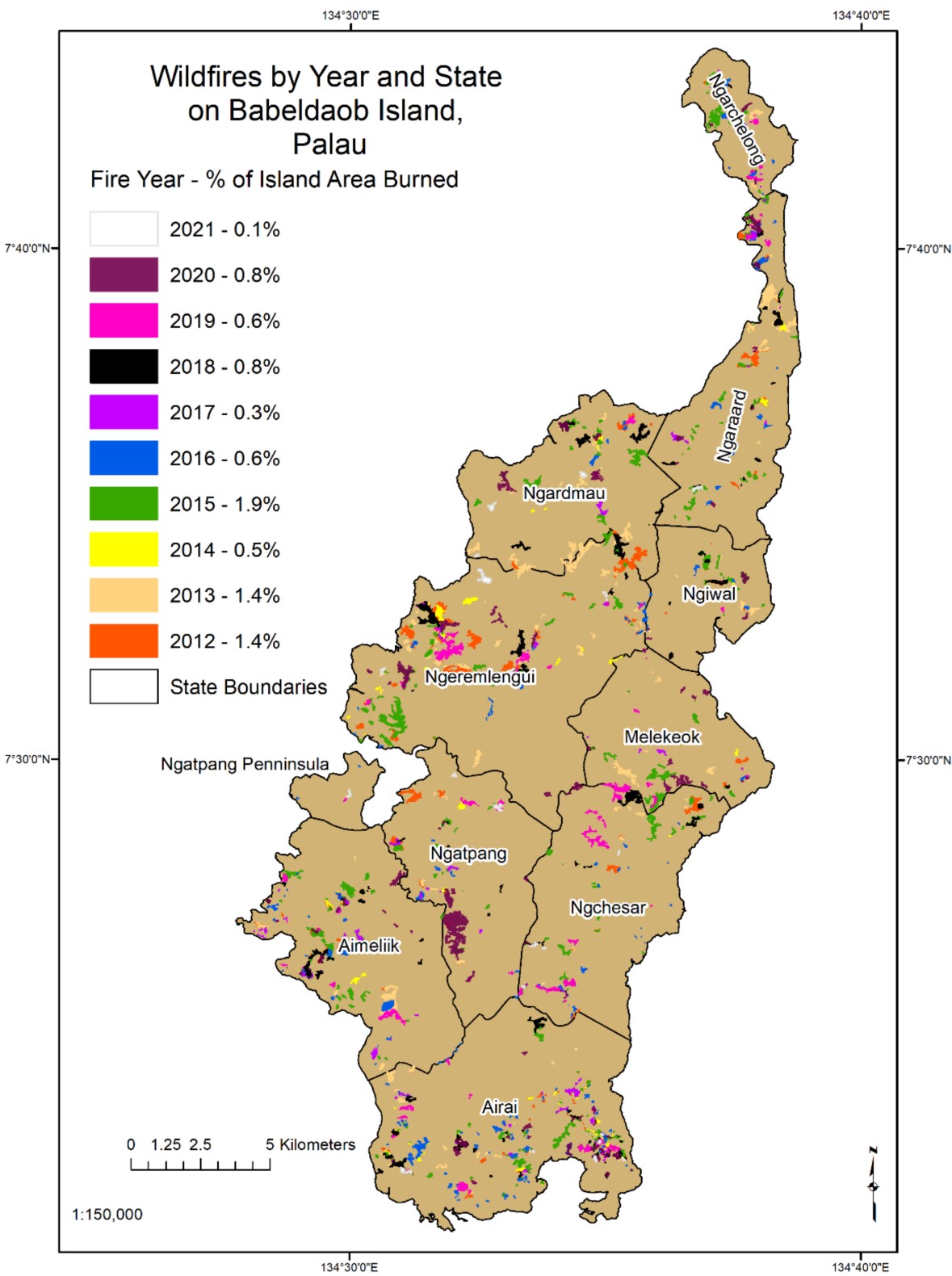

**Figure 3.** Annual wildfires and percentage of island area burned on Babeldaob Island, from 2012 to 2021. Some older burned areas are obscured by more recent burned areas.



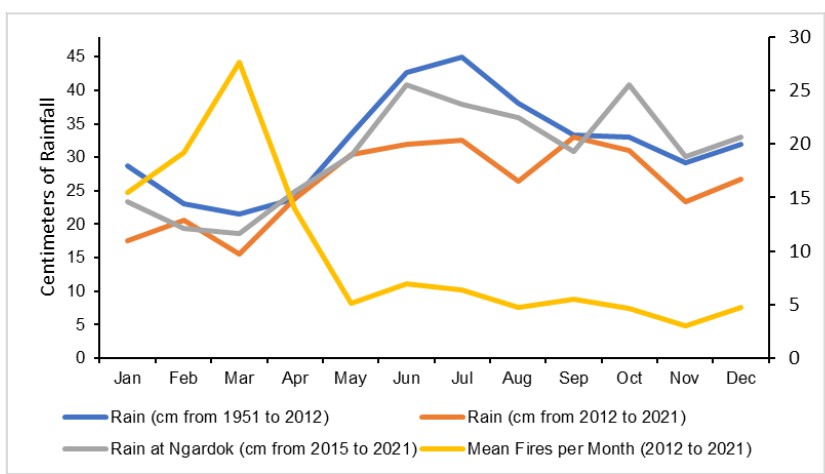

**Figure 4.** Mean monthly wildfires and rainfall (cm) on Babeldaob Island, Palau, showing fires that could be assigned a fire date to the nearest month, rainfall at the meteorological station on Koror Island for the available historical period before and during the study (data from 2013 were removed due to incompleteness), and rainfall for 2015–2021 at the meteorological station at Lake Ngardok Nature Reserve in central Babeldaob.

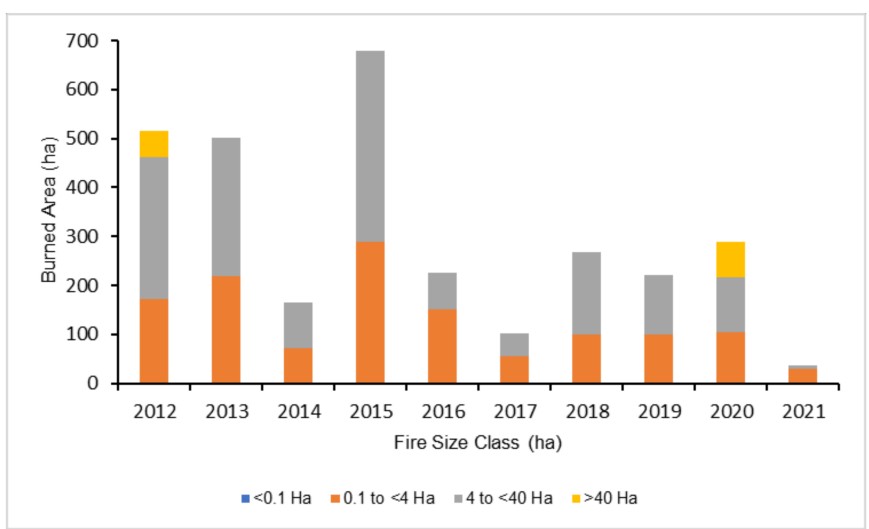

**Figure 5.** Wildfire burned area (ha) per year from 2012 to 2021 on Babeldaob Island, Palau, by National Wildfire Coordinating Group size classes.

**Table 5.** Number and percentage of wildfires, area burned (ha), and percentage of area burned on Babeldaob Island, Palau, from 2012 to 2015, by cause of ignition.

| Cause of Ignition | # of Fires | % of Fires | Area Burned (ha) | % Area Burned |
|---|---|---|---|---|
| Hunters | 80 | 21% | 294 | 33.6% |
| Arson | 129 | 33.9% | 270 | 30.8% |
| Farm | 113 | 30% | 191 | 21.8% |
| Government | 26 | 6.8% | 70 | 8% |
| Clear Vegetation | 10 | 2.6% | 26 | 2.9% |
| Brush Pile | 14 | 3.7% | 18 | 2.1% |
| House Building | 5 | 1.3% | 6 | 0.7% |
| Other | 2 | 0.5% | 0.3 | 0.04% |

Wildfires occurred across Babeldaob Island, although wildfire statistics varied strongly by state and trends by geospatial proximities (Tables 6 and 7). The states of Ngaremlengui, Airai, and Aimeliik had the most fires and burned area (including forest) per year, which were four to five times greater than Melekeok and Ngiwal states, which had the least. Over one-half of all fires occurred within 50 m of roads, and 6–7% of fires occurred within 10 m of streams and mangroves; only 1% occurred within 10 m of the coastline.

**Table 6.** Number of wildfires, size range (ha), mean area (ha), sum of burned area (ha), and percentage of state area burned on Babeldaob Island, Palau, from 2012 to 2021 by state.

| State Name | # of Fires | Size Range (ha) | Mean Area (ha) | Sum of Area (ha) | % of State Area |
|---|---|---|---|---|---|
| Aimeliik | 214 | <0.1–28 | 2.0 | 423 | 10.2% |
| Airai | 474 | <0.1–21 | 1.1 | 533 | 9.7% |
| Ngchesar | 121 | <0.1–19 | 1.8 | 218 | 5.3% |
| Ngatpang | 71 | <0.1–73 | 3.3 | 234 | 5.9% |
| Melekeok | 58 | <0.1–17 | 1.9 | 113 | 4.4% |
| Ngiwal | 56 | <0.1–12 | 1.7 | 96 | 5.5% |
| Ngaremlengui | 214 | <0.1–53 | 3.1 | 668 | 10.3% |
| Ngardmau | 82 | <0.1–17 | 3.7 | 300 | 8.9 |
| Ngaraard | 136 | <0.1–24 | 2.2 | 299 | 9.2 |
| Ngarchelong | 70 | <0.1–14 | 1.3 | 94 | 8.4% |

**Table 7.** Number and percentage of total wildfires, mean (ha), sum (ha), and percentage of total burned area by geospatial category on Babeldaob Island, Palau, from 2012 to 2021.

| Geospatial Category | # of Fires | % of Fires | Mean Area (ha) | Sum of Area (ha) | % of Burned Area |
|---|---|---|---|---|---|
| W/in 500 m of roads | 1250 | 83.6% | 2.0 | 2498 | 83.9% |
| W/in 50 m of roads | 798 | 53.3% | 2.1 | 1695 | 56.9% |
| W/in 50 m of streams | 373 | 24.9% | 3.5 | 1296 | 43.5% |
| W/in 10 m of streams | 98 | 6.6% | 4.1 | 401 | 13.5% |
| W/in 10 m of mangrove | 94 | 6.3% | 3.6 | 337 | 11.3% |
| W/in 50 m of coastline | 34 | 2.3% | 3.7 | 125 | 4.2% |
| W/in 10 m of coastline | 16 | 1.1% | 4.7 | 76 | 2.5% |

Over 28% of all GPS-mapped fires (2012–2015) included observations of one or more invasive species, including: *Chromolaena odorata* ((L.) R.M.King & H.Rob)., *Bothriochloa bladhii* ((Retz.) S.T.Blake), *Imperata* sp., *Mimosa* sp., *Stachytarpheta* sp., *Adenanthera pavonina* (L.), *Leucaena leucocephalla* ((Lam.) de Wit), and *Acacia auriculiformis* (Benth). Invasive species were recorded in all states of Babeldaob, but rates were higher in Aimeliik and Airai states. *Chromolaena odorata* was the most common invasive species recorded in burned areas, mostly in grassland or abandoned agricultural areas. It has been described as a species that burns easily and returns quickly after burning, resulting in a cycle that leads to more fire-prone vegetation. Despite being infected with a biocontrol agent, this weed is still one of the most common invasive species on Babeldaob [24]. *Imperata* sp. grass is of extreme concern in terms of invasive and wildfire potential and is under observation by Palau's national government but was only observed in a few small patches in Airai state and one small patch in Melekeok state. *Acacia auriculiformis* has become naturalized on Babeldaob and is not currently considered to be a serious invasive species, although burned areas close to mature trees were often observed with dense seedling regeneration.

About 27% of total burned area burned at least a second time, and repeatedly burned areas occurred in every state of Babeldaob (Figure 6). These reburned areas had about

half as much forest area burned proportionally compared with total burned area (7% vs. 14%), and proportionally, more savanna burned (91% compared with 83%). About 4.5% of the combined burned area of any two fire years was attributed to repeated fires, with the highest proportion of reburns occurring between the two fire years of 2012 and 2018 (16%), and the largest total area of reburn occurring between 2012 and 2015 (131 ha). The percentage of burned area burning three times over the study period reached a maximum of 2.8% of total burned area for the 2012, 2014, and 2017 fire year combination.

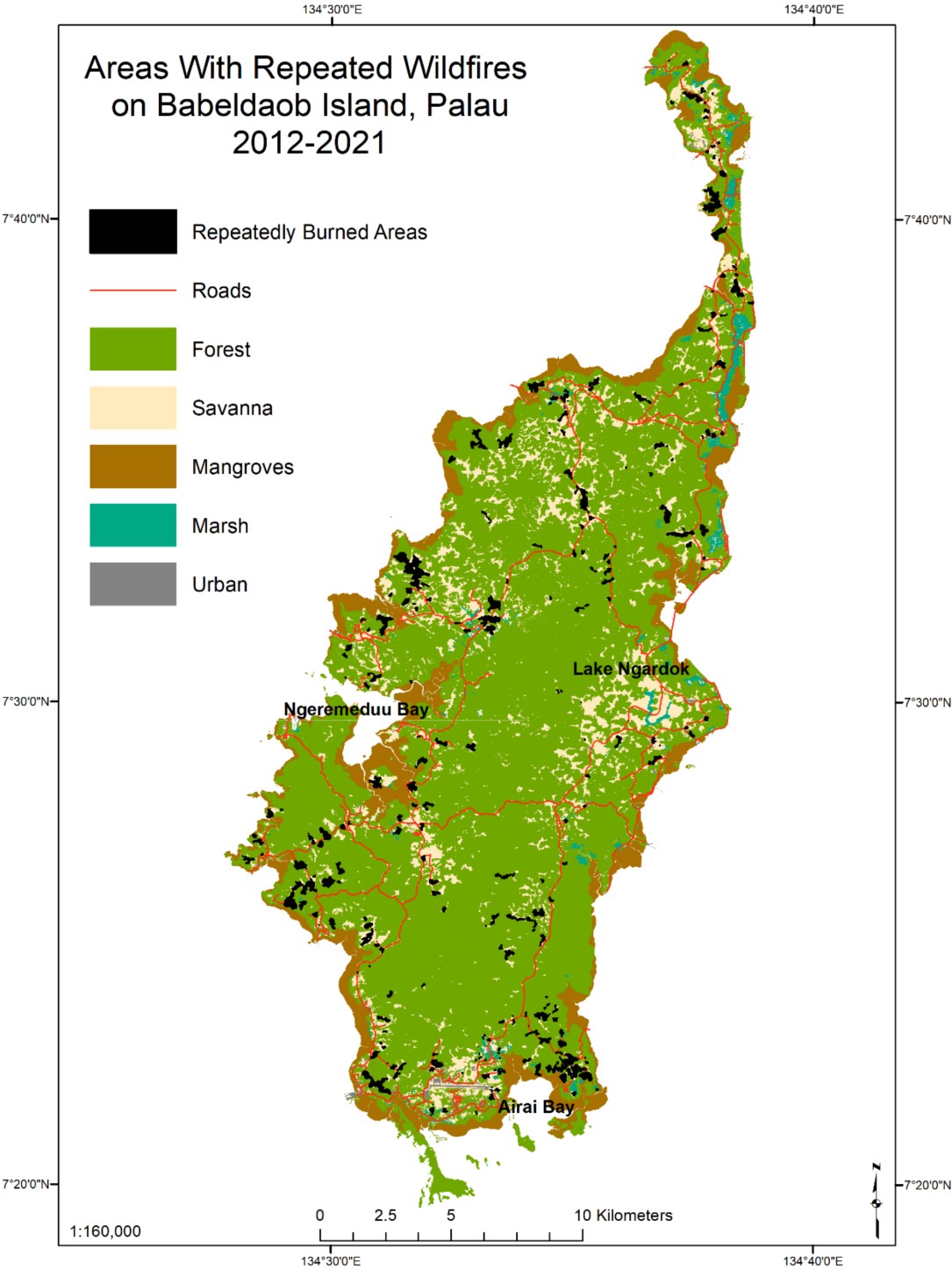

**Figure 6.** Areas with repeated wildfires from 2012 to 2021 on Babeldaob Island, Palau, and coverage of major vegetation classes.

## 5. Discussion

The 2012 to 2021 fire history presented here represents the most robust documentation of wildfire activity in Micronesia to date and provides the first accurate estimates of wildfire occurrence on Babeldaob Island, Palau. Three distinct methods were incorporated; however, they overlapped and had very similar burned area values, and their combination from 2012 to 2015 likely resulted in slightly more thorough coverage of burned area on Babeldaob than the period 2016–2021, when satellite imagery was exclusively utilized. Despite the excellent coverage of imagery available through the DGEV web service, cloud cover was an issue and obscured some burned areas. The extent of cloud cover varies from year to year, and most burned areas remain visible for several months, but grassland vegetation after fires in Aimeliik and Airai states can recover in less than one month; thus, it is possible that we underestimated burned area where grass recovery is rapid, particularly during the second half of the study.

The ground-based measurements between 2012 and 2015 allowed for the documentation of wildfire causes, vegetation types burned, depth of forest edge affected, and the presence of invasive species, data that cannot be collected with remote sensing, at least not at such a fine scale. However, maintaining the field surveys over time was difficult, and some burned areas were remote or difficult to see. Additional variation likely resulted from differences in surveyor experience and where steep terrain or dense vegetation could cause surveyors to stray from the burned area boundary. Categorizing wildfire causes was imprecise when the cause was not obvious, or when more than three months had elapsed since the burn event. Overall, however, the approach was evidence-based, and the results aligned closely with a household fire survey on Babeldoab Island [42]. Determining the time since the burn, as well as the exact boundary being surveyed, was difficult because rates of vegetation recovery varied substantially from site to site. In addition, if we had been limited to handheld GPS and bi-annual aerial photo documentation, we would have underestimated burned area and likely temporally misattributed a small proportion of fires. The intensity of ground-based efforts and experience gained from 2012 to 2015, as well as the timing of cloud-free windows for aerial photo collections, generally determined the proportion of burned area requiring supplementation and/or correction with archival satellite imagery, with 56%, 13%, 48%, and 2% of burned area supplemented by respective year. The satellite-based approach required access to imagery but could be accomplished quickly and without fieldwork. Burned area estimations of larger islands or landscapes would be better addressed through applications of a spectral index such as the NBR; however, we have found that utilizing the DGEV web service and manual digitization of burned areas on small tropical islands is equally efficient and minimizes errors of omission [20]. However, the NBR also offers estimations of fire severity, which we did not address at all, and which are very difficult to objectively estimate visually in the field, and thus could be a topic of further research in relation to fire cause or topography.

This burned area inventory allowed for Palau's first spatial and temporal analyses of wildfires. Our documentation can provide a rough blueprint for how a small country can establish a landscape-scale regional fire history relatively quickly through utilizing low-tech, low-cost methods and leveraging partnerships to access high-resolution remote sensing resources. Once an inventory is available, it can support resource manager efforts, such as fire-focused decision support tools [43], the development of national wildfire protection plans, the identification of areas burned repeatedly, and understanding the impacts of wildfire on conservation resources. This relatively rare tropical case study could also be useful for making comparisons with other regional and international wildfire datasets, particularly on small tropical islands with savanna habitats.

Our results confirm environmental and public safety concerns about wildfire threats on Babeldaob Island. In the past 10 years, we observed several destroyed structures, threatened power grid infrastructure, numerous post-fire rain-driven landslides, strong evidence of post-fire erosion, and fires that burned into wetlands or to the edge of steep coastlines. The damage and destruction to personal and public agricultural and agroforestry

resources, such as coconut and mahogany groves or taro patches, was also common. Here, we summarize the geospatial characteristics, climate-related dynamics, and potential cumulative impacts of this anthropogenic threat to Babeldaob's natural resources.

### 5.1. Wildfire Size and Burned Land Area

The substantially higher number and larger footprint of small fires (<4 ha) compared with large fires (>4 ha) on Babeldaob aligns with general patterns of other islands in Western Micronesia [44] but differs from Hawai'i where most burned area is attributed to large fires [15]. The predominant vegetation type burned in Palau was savanna (83%), with half being savanna fern lands—similar to the burned area dominance of fire adapted grassland vegetation in Hawai'i (80%) [13]. The proportion of land area burned on Babeldaob was higher on average than what is observed in the western United States or Hawai'i [15]. The two largest fires (53 and 73 ha) observed during the study occurred in the typically driest months (February 2012 and March 2020), but in two average moisture years. In contrast, 2015, an El Niño year, saw the most fires and largest burned area statistics.

### 5.2. Forest Edge Effects of Wildfire

Although field observations indicate that forest edge on Babeldaob was substantially impacted from fires, the actual damage was difficult to precisely quantify. The clipped burned area of the 2014 vegetation map indicated the ~1.1% of total forest area had burned over the study period, or ~ 0.1% per year on average, which did not account for other forms of forest loss or forest gain between 2012 and 2021. Although 43% of fires with field observations (2012–2015) indicated burned forest edge, fires were patchy, and the depth of forest edge impact varied. The forest edge is dense with vegetation and is much moister than the adjacent savanna habitat, with fires generally not moving more than 5 m into forest habitat from savanna. During dry and windy conditions, however, large trees inside the forest edge were damaged, with fires moving deeper into forest habitat. We observed fire damage 8 m or more inside the forest edge on seven occasions, all during the 2015 drought. However, because most species of trees and shrubs in savanna and forest edge habitats are partially resistant to fire and/or can resprout after being burned, individual trees would need to be monitored over time after being burned to determine mortality rates [45,46].

### 5.3. Ecological and Landscape Trends of Wildfire

Post-fire recovery on Babeldaob Island shows significant variation in recovery rates among savanna sites. In Palau, there are two savanna plant species assemblages, one occurring on upland volcanic soils and the other on marine terrace soils [46,47]. Based on this preliminary study of post-fire vegetation recovery rates, and our own field observations, non-woody vegetation recovery occurs faster on marine terrace soils supporting the grassland savanna subtype. The majority (65%) of the most common savanna plant species in recovering areas were observed to exhibit re-sprouting ability after fires [46]. In addition, small patches of forest vegetation often survive burning, particularly within fern land and shrub savanna types, and can persist for decades as remnant forest fragments within a savanna matrix [8].

Landscape-scale studies of Babeldaob have shown unassisted forest recovery after disturbance, particularly next to forest edges and abandoned agricultural areas [48,49]. This can happen quickly, as forest cover increased by ~7% (61.7 to 68.2%) in the 30 years after WWII, and another ~4% (68.2 to 71.9%) in the 30 subsequent years [50], but rates of recovery varied with the intensity and duration of land use during the Japanese colonial period and commercial administration of the island from 1921 to 1945 [49]. Forest patches in degraded areas recover more slowly; even inside a protected area without wildfires the natural expansion rate was only ~2 cm laterally per year [8]. An island-wide aerial photo vegetation comparison of Babeldaob indicated ~4.1% of forest area lost, and 3.5% of forest area gained (~0.6% net loss per year) from 1976 to 1992 [50]; thus, Babeldaob appears to have a stable distribution of forest and savanna that was likely maintained through fire

activity over this time period. In the two decades since 1992, forest cover appears to have increased slightly (~0.25% per year) [50].

### 5.4. Climactic Drivers of Wildfire Patterns

Wildfires on Babeldaob were most common from January through April, during which mean monthly rainfall is typically 30% less than annual average monthly precipitation [25]. Sustained drier periods result in more and larger wildfires, and the driest month of March had six times the monthly mean number of fires per island as months from May to December (28 to 5). Dry and windy conditions can lead to bursts of burning activity, even during wetter times of the year, and average rainfall during the study may have been less than during the recent historical period before the study (Figure 4).

Internationally, 2015–2016 was associated with unusually dry and warm conditions, along with increased fire activity and massive net emissions of carbon dioxide across the entire pantropical region. In one three-month period, tropical peatland fires in Indonesia released more than 1 million tons of $CO_2$ [2]. Longer stretches of hotter and drier weather during El Niño-related droughts led to a greater number of larger fires in Palau and in other Pacific island nations [15,51]. In 2015, fires burned the largest overall area and proportion of Babeldaob Island during the ten-year study. Based on diverse indicators and historical standards, 2015–2016 was a relatively severe El Niño year. Drawing from 80 years of weather data, severe drought appears to be consistently associated with El Niño events in Palau, which occur about every 10–15 years [51]. This is not the case with the Pacific region overall, where not all El Niño events result in drought, and effects from El Niño can differ among eastern, central, and western regions [51].

Compared with wetter and more minor fire years such as 2017, the total burned area in 2015 was 6.6 times larger, whereas the average fire size was 4.1 times larger than in 2017. Interestingly, the two largest fires on record did not occur in 2015. Fires occurred in all states of Babeldaob in 2015–2016, and there was no difference among states for that year compared with patterns observed over the study period.

### 5.5. Reasons for Ignitions

Fires hunter–gatherers set to facilitate access to hunting and gathering areas resulted in the marginally highest total percentage area burned by category (34%). Although we did not directly measure fire intensity, longer time intervals between these burns compared with roadside areas (allowing for greater biomass accumulation, typically in the form of the fern *Dicranopteris linearis* ((Burm. f.) Underw.)), and more obvious burn scars (visible from aerial photos and satellite imagery for three years plus), point to hunting–gathering fires being more severe than other categories of fire.

Arson caused the second largest percentage of burned area across categories (31%). Fires next to roads with additional nearby spot fires were identified as arson, the majority of which (58%) occurred within 50 m of roads. Most fires attributed to government (national and state) causes were also on or near roads and were generally associated with roadside vegetation maintenance. It cannot be ruled out that some of these fires were caused by arsonists lighting fires soon after maintenance work had finished. It is unknown how improved road access over the past 15 years has influenced the numbers and impacts of arson and hunting–gathering fires. However, given the small increase in forest cover on Babeldaob Island over the past 30 years, the widely observed tropical pattern of road building increasing fires and land clearing might not apply in Palau.

Farm fires contributed about 22% of total area burned and were the most common reason for burning. Fire is useful for clearing fast-growing tropical vegetation that competes with food plants, covers land boundaries or markers, or accumulates around structures; this use of fire is common throughout the year on Babeldaob. Similar to brush piles and land clearing activities, farm fires often start as gathered piles of vegetation left to smolder while people attend to other chores, and rarely escape control except under dry and windy conditions. However, uncontrolled fire used to clear vegetation before planting can create

burned areas than that are much larger than the eventually planted area. In addition, most nutrients released by burning can be quickly lost to rain-driven runoff [52].

### 5.6. Watershed Impacts of Wildfires

Fires are likely impacting many features of the environment in and around Babeldaob, with at least 14% of total area burned over ten years being within 10 m of streams, 11% within 10 m of mangroves, and 3% within 10 m of coastline. These proximities highlight the impact of fire on erosional process and resulting sediment delivery to Babeldaob's streams, mangroves, and near-shore marine areas including sea grass and coral reef habitats. Sedimentation on Babeldaob reduces coral cover and diversity, and peak sedimentation rates have been attributed to land development [38,39]. A hydrological-modeling-based watershed decision support tool for Babeldaob predicted the highest sediment yields in catchments with little forest cover, with sediment hotspots primarily being located near coastal population centers in Airai and Melekeok States, and the basins surrounding Ngeremeduu Bay [43]. In a study of the Ngerikiil watershed, an important water source for Palau but also a sediment source for Airai Bay, unpaved roads were found to be the highest sediment-generating source (175,119 kg/ha/year), followed by farm tilling (109,403 kg/ha/year), savanna burning (36,713 kg/ha/year), and unprotected construction activities (27,902 kg/ha/year) [52]. On Guam, post-burn sedimentation rates were among the highest reported in the literature, particularly at the onset of the wet season, and the presence of burning and badlands increased soil loss by 35% compared with areas without burning and badland restored to savanna [37]. Sedimentation and loss of forest canopy due to road building was shown to adversely affect physical and biological characteristics of streams on Babeldaob, and there is a need for further research in the Pacific on specific widths of riparian forest buffers to protect ecosystem services [53].

Approximately 11% of total non-forest vegetation on Babeldaob burned twice or more over ten years. Sites that burn repeatedly over short time periods are at higher risk of biodiversity loss, accelerated erosion, and vulnerability to non-native plant species invasion, and our mapping of repeat fires can support efforts to target these areas for outreach or management [37]. Repeatedly burned areas occurred in every state of Babeldaob, in locations previously identified as threatened by wildfires [24]. These include several areas in states with the highest average burn metrics overall (including Airai and Ngeremlengui), but the study also highlighted many areas in Aimeliik state and remote areas within protected areas in Ngeremlengui state being burned repeatedly.

### 5.7. Management and Prevention of Wildfires

Based on data from Palau Fire and Rescue Division, from 2007 to 2009, there were 48 total fires which burned about 7 ha over the whole island of Babeldaob [24]. In comparison, even during the least severe fire year of the study in 2021, we documented five times more than that total burned area in a single year. The much smaller estimates likely relate to how the Fire and Rescue Division documented these fires, which only included visual area estimates of fires that were viewed as a threat to infrastructure.

Generally, wildfires are not suppressed, but rather left to burn themselves out over a period of hours to days. This is partly because the steep, hilly terrain of Babeldaob Island and the remote location of many of the largest fires create challenging conditions for wildfire suppression. This study revealed that the landscape-scale impacts of wildfires on Babeldaob Island have been substantially underestimated and highlights the role of public education in wildfire prevention, with geospatial wildfire trends over time being used to highlight the most affected areas needing agency and community attention.

The largest wildfire documented in Palau occurred in 2020. In response, stakeholders from across Palau in the national government, state governments, and non-governmental organizations formed the Network for Wildfire Prevention (NWP). The NWP has focused on capacity building of stakeholders on wildfire issues, dry season wildfire response planning, and public outreach to raise awareness on the impacts and policies related to

burning and wildfires. When the NWP was formed in 2020, there were 109 fires that burned 289 hectares across Babeldaob Island. There was a decline in wildfires in 2021 to 29 fires covering 38 hectares, potentially attributable to the efforts of the NWP. Wildfire history mapping remains critical to inform the NWP of the current wildfire situation and how fire events compare to those of previous years.

## 6. Conclusions

This ten-year study was a collaborative effort to collect wildfire history data on Babeldaob Island, Palau, in order to inform the Nation's natural resource management and conservation efforts. Our study questions were: (1) What is the annual area burned; (2) What are the spatial and temporal patterns of wildfires; (3) What is the level of wildfire threat to forest resources; and (4) What are the principal causes of wildfire ignitions on Babeldaob Island? We used three overlapping methods of mapping burned areas to arrive at these answers: (1) the mean area burned was about 300 hectares or 0.8% of the total island area (including mangroves); (2) wildfires occur across the entire island in all states, with some states seeing much higher burned areas than others, and in all vegetation types but primarily within savanna habitat during dry season months; (3) wildfires present a minor threat to forest resources, burning only about 0.1% of forest area per year; and (4) hunting–gathering, arson, and farming activities are the principal causes of ignitions on Babeldaob. Wildfires clearly perpetuate savanna habitat on the island, with a high proportion of savanna plant species displaying pyrophytic tendencies, and savanna was highly variable across the island in terms of vegetation cover and rate of recovery after burning. We found that the high-resolution imagery provided by the DGEV online image service is well suited for the efficient monitoring and mapping of burned areas on small tropical islands in Micronesia but utilizing overlapping local and low-cost methods such as handheld GPS mapping can add important information regarding environmental impacts of wildfires at a fine scale.

**Author Contributions:** Conceptualization, J.D., S.C., C.P.G., D.M. and P.L.C.; methodology, J.D., D.M. and P.L.C.; software, J.D.; validation, J.D.; formal analysis, J.D.; investigation, J.D., D.M., P.L.C. and T.H.; resources, J.D., P.L.C., S.C., C.P.G. and A.U.; data curation, J.D.; writing—original draft preparation, J.D.; writing—review and editing, J.D., P.L.C., S.C., C.P.G., A.U. and T.H.; visualization, J.D.; supervision, S.C., C.P.G. and A.U.; project administration, J.D., S.C., C.P.G. and A.U.; funding acquisition, S.C., C.P.G. and A.U. All authors have read and agreed to the published version of the manuscript.

**Funding:** This research received no external funding.

**Institutional Review Board Statement:** Not applicable.

**Informed Consent Statement:** Not applicable.

**Data Availability Statement:** Not applicable.

**Acknowledgments:** We thank the USDA Forest Service, including Region 5 and the Pacific Southwest Research Station, Institute of Pacific Islands Forestry in Hilo, Hawai'i, for providing funding support that made this project possible. We thank Palau Forestry staff for assisting with field-based mapping. We also thank Melekeok Conservation Network staff, who were important partners in reporting fires and walking fire perimeters. We thank Palau's Protected Area Network, which established a Fire and Invasive Species Coordinator, Isechal Remengesau, who maintained the effort to monitor and map fires on Babeldaob for two years to document the trends of fire on the landscape. We thank Jolie Liston for assistance with literature review.

**Conflicts of Interest:** The authors declare no conflict of interest.

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
