# Peer review of "Dynamics of Anthropogenic Wildfire on Babeldaob Island (Palau) as Revealed by Fire History"

_fire, doi:10.3390/fire5020045_

Round 1

Reviewer 1 Report

Review on Ms. Ref. No.: fire-1654192

Dynamics of anthropogenic wildfire on Babeldaob Island, Palau as revealed by fire history

Julian Dendy, Dino Mesubed, Patrick Colin, Christian Giardina, Susan Cordell, Tarita Holm, Amanda Uowolo

Recommendation: Accept after major revision

  1. Summary

Presented research assesses the potentially growing threat of wildfires, where the authors mapped wildland fires on Babeldaob Island using ground-based surveys and aerial photographs between 2012 and 2015, and satellite imagery between 2012 and 2021.

  1. Major issues
  • The whole paper looks like a technical/statistical report of a mission abroad rather than a research paper. Therefore I can’t get what are the main gap and the main benefits of your work. I don’t say that they are missing, just not clearly explained, and difficult to guess your thoughts (maybe lines 344-360 need great expansion and better clarification).
  • Lines 20-21: At first look, I am wondering why you have two assessment periods - 2012-2015 and 2012-2021. Would you like to explain the cause?
  • Everywhere in the text: Please take a look at the Instructions for authors on the Fire website and revise the style of the references accordingly. The authors’ names and year of the reference are not required in the main text of the paper, [1-2] is enough. Also, revise the reference list style and include DOIs to all research papers available, without [Verified …].
  • Lines 85-89: The Introduction usually concludes with the plan for the rest of the paper. This plan should not include results only the plan of what will be presented.
  • A Conclusion section is missing. Every paper shall have a conclusion.
  1. Minor issues
  • Line 225: Where does this map come from? Reference?
  • Do not count and refer to papers, which are “in preparation” phase. References 41,47.
  1. Opinion

I assess the overall manuscript as an average-level paper and I recommend it for possible acceptance after major revision related to my notes above.

Kind regards!

Author Response

  1. I suppose I see your point, and we will attempt to better explain the gaps and benefits of the work, including expansion and clarification of the discussion introduction. We do mention some of the potential gaps in the work, like the lack of fire severity assessment, but there are more that can be mentioned or further elaborated. The main benefit is that we addressed a national level data gap and request “ by comprehensively quantifying the spatial extent, temporal variability, and ecological and social characteristics of fire on Babeldaob Island from 2012 to 2021.”(lines 81-83) These characteristics were previously unknown and unstudied, and the landscape scale, relatively long time period, and tropical island location give it relative importance for regional and habitat-based comparisons.
  2. When we began the study, we didn’t have access to satellite imagery, so the main focus was to use ground-based mapping using handheld GPS and supplement that with biannual aerial photo collection. This was also when we collected field observations on vegetation burned, depth of forest edge affected, invasives species present, and reasons for ignitions, as we mention in lines 21-23, “Data on causal factors, vegetation type, and presence of invasive species were collected between 2012 and 2015, with hunting, arson, and agricultural clearing being the principal causes of ignitions.” After we gained access to online satellite imagery including archival imagery, we were able to go back and double check our field work, as we mention in lines 202-205 “We used publicly available satellite images of Babeldaob to supplement the 2014 and 2015 layers and used most of the available cloud-free archival imagery in the DGEV imagery service to check, correct, and supplement fire layers from 2012 to 2015.” The ground-based field work is difficult to maintain over time and the satellite archive-based method is considerably more efficient, so we transitioned to this method exclusively in 2016. So, we had one time period with ground-based observations incorporated (2012-2015), and the overall time period of the study (2012-2021) which used satellite imagery. Perhaps we should omit mention of this from the abstract?

  3. I will change the reference style to fit the journal, but I was following these instructions in the Instruction for Authors page on the Fire website, “Your references may be in any style, provided that you use the consistent formatting throughout. It is essential to include author(s) name(s), journal or book title, article or chapter title (where required), year of publication, volume and issue (where appropriate) and pagination. DOI numbers (Digital Object Identifier) are not mandatory but highly encouraged”

  4. I agree with you here, but I was following these instructions in the Introduction section of the Instructions for Authors page on the Fire website, “Introduction: The introduction should briefly place the study in a broad context and highlight why it is important. It should define the purpose of the work and its significance, including specific hypotheses being tested. The current state of the research field should be reviewed carefully and key publications cited. Please highlight controversial and diverging hypotheses when necessary. Finally, briefly mention the main aim of the work and highlight the main conclusions. Keep the introduction comprehensible to scientists working outside the topic of the paper.”

  5. I would be happy to include a Conclusion section, but I was following these instructions from the Conclusion section of the Instructions for Authors page on the Fire website,” Conclusions: This section is not mandatory but can be added to the manuscript if the discussion is unusually long or complex.”

  6. I created this map, and it is referenced in lines 212-213 in the first sentence of the Results section, “Between 2012 to 2021 we documented and mapped at least one fire every year in every state on Babeldaob Island, Palau, except in 2021 (Table 1 and Figure 1).”

  7. I will remove or change these references if necessary, but I was following these instructions from the Unpublished Data section of the Instructions for Authors page on the Fire website, "Unpublished data" intended for publication in a manuscript that is either planned, "in preparation" or "submitted" but not yet accepted, should be cited in the text and a reference should be added in the References section.”

Reviewer 2 Report

The paper proposed Dynamics of anthropogenic wildfire on Babeldaob Island, Palau as revealed by fire history. Experimental results show the good performance of the proposed method. However, some issues should be addressed before the publication.

Major issues:

1) The structure of the paper is not standardized. I think the part of Section 2 should be divided into two parts: data introduction and method introduction. The flow chart of the proposed method should be given.

2) Only one set of experimental data is used in this paper, which cannot generally prove the advantages of the proposed method. It is recommended to add at least one more set of experimental data.

Minor issues:

1) The methods proposed by the authors is mainly based on remote sensing image. In the introduction part, a comprehensive and systematic background introduction and description related to the advanced and latest works, e.g.,

[1] Registration of Multiresolution Remote Sensing Images Based on L2-Siamese Model, IEEE Journal of Selected Topics in Applied Earth Observations and Remote Sensing, 2021, 14: 237 - 248.

[2] Super-Resolution Mapping Based on Spatial-Spectral Correlation for Spectral Imagery [J]. IEEE Transactions on Geoscience and Remote Sensing, 2021, 59(3): 2256-2268.

2) The last part of the introduction needs to give chapter arrangement.

Author Response

  1. I was not aware that the methods section should be standardized in this way; it’s not mentioned in the Instructions for Authors.  I am open to restructuring as you recommend and including a flow chart of the proposed method.
  2. We don’t have any experimental data since we didn’t intentionally set any fires. We present observational geospatial data of wildfires, and use several additional geospatial datasets (roads, coastline, vegetation cover, state boundaries, and protected area boundaries) to summarize the spatial extent and temporal variability of fire on Babeldaob Island from 2012 to 2021, as well as ecological and social characteristics of fires from 2012 to 2015.  We provide the estimated accuracy of our methods and some commentary on pros and cons of each in the context of mapping wildfires on a relatively small landscape scale, so we did not intend to prove the advantages of any methodology, but rather utilize whatever methods were available to us to address the national level data gap and request. 

  3. This point is well taken, and I appreciate the reference examples you have shared here.  I will attempt to incorporate these and other sources to present a paragraph on background introduction related to recent remote sensing work.

  4. This is not mentioned in the Instructions for Authors page, but I will include a chapter arrangement in the introduction. 

Reviewer 3 Report

The article is well written and actually is ready for publication.

What is not clear:

(a) the authors used three methods of survey and data analysis (GPS, aerial photo and satellite) that give strongly different results (Table 7). I did not find in M&M and the Results (tables 1...6) what  method (GPS or aerial photo) was used for the period 2012-2015 that can significantly influence on the results.

(b)  The Table 7 itself should be in the M&M

Author Response

  1. I will attempt to clarify this further in the manuscript, but the handheld GPS method used from 2012-2015 was the only method we used which made it possible to collect observations on vegetation cover type burned, depth of forest edge penetrated, invasive species present, and reasons for ignitions.  Table 7 does not indicate different results; it indicates the number of fires and the percent of burned area documented by each of the three methods per year from 2012-2015.
  2. This is a good idea and if the journal format allows for it then I will move Table 7 to the methods section.

Round 2

Reviewer 1 Report

I appreciate the author's effords in improving the paper. Really good job! I see that my notes and recommendations are taken into account and I have no more comments. Therefore I recommend the paper for acceptance in the present revised form!

Congratulations and best regards!

Author Response

Dear reviewer #1,

Thank you so much for your time, attention, and warm words.  

They are all much appreciated. 

Reviewer 2 Report

Thanks for the authors' reply. I have no other problems here.

Author Response

Dear Reviewer #2,

I extend a tremendous thank you for your time and attention in helping us get this manuscript in the best possible shape to share with the public.